# Sustainable Pickering Emulsions with Nanocellulose: Innovations and Challenges

**DOI:** 10.3390/foods12193599

**Published:** 2023-09-28

**Authors:** João Paulo Saraiva Morais, Morsyleide de Freitas Rosa, Edy Sousa de Brito, Henriette Monteiro Cordeiro de Azeredo, Maria Cléa Brito de Figueirêdo

**Affiliations:** 1Embrapa Cotton, Rua Oswaldo Cruz 1143, Campina Grande 58428-095, Brazil; joao.morais@embrapa.br; 2Embrapa Tropical Agroindustry, Rua Dra. Sara Mesquita 2270, Fortaleza 60511-110, Brazil; clea.figueiredo@embrapa.br; 3Embrapa Food and Territories, Rua Cincinato Pinto, 348, Maceió 57020-050, Brazil; edy.brito@embrapa.br; 4Embrapa Instrumentation, Rua XV de Novembro 1452, São Carlos 13560-970, Brazil; henriette.azeredo@embrapa.br

**Keywords:** sustainability, Pickering emulsions, nanocellulose, regulatory issues

## Abstract

The proper mix of nanocellulose to a dispersion of polar and nonpolar liquids creates emulsions stabilized by finely divided solids (instead of tensoactive chemicals) named Pickering emulsions. These mixtures can be engineered to develop new food products with innovative functions, potentially more eco-friendly characteristics, and reduced risks to consumers. Although cellulose-based Pickering emulsion preparation is an exciting approach to creating new food products, there are many legal, technical, environmental, and economic gaps to be filled through research. The diversity of different types of nanocellulose makes it difficult to perform long-term studies on workers’ occupational health, cytotoxicity for consumers, and environmental impacts. This review aims to identify some of these gaps and outline potential topics for future research and cooperation. Pickering emulsion research is still concentrated in a few countries, especially developed and emerging countries, with low levels of participation from Asian and African nations. There is a need for the development of scaling-up technologies to allow for the production of kilograms or liters per hour of products. More research is needed on the sustainability and eco-design of products. Finally, countries must approve a regulatory framework that allows for food products with Pickering emulsions to be put on the market.

## 1. Introduction

Pickering emulsions are garnering more attention in the food sector [1,2,3]. The stabilizing mechanism of a Pickering system is different from the mechanism commonly observed for traditional emulsions, formed by immiscible liquids like oil and water [4]. Most conventional emulsions use some surfactant or tensoactive chemical as stabilizer, reducing the surface tension between the polar and nonpolar phases. Pickering emulsions, on the other hand, are formed by solid stabilizing particles with polar and nonpolar domains that act as a physical barrier between the phases, preventing the coalescence of droplets. A chemical coats the interface between the droplet (dispersed phase) and the dispersing medium (continuous phase), with no solubilization in any of the phases [5], usually forming a single layer of solid particles. When the contact angle of one of these single particles towards the water phase lies between 15° and 90°, an oil-in-water Pickering emulsion forms, and if the contact angle ranges from 90° to 165°, it stabilizes the water-in-oil emulsion [6]. If the solid particle size with adequate contact angle ranges between 10 and 20 nm, the energy barrier prevents the separation between the droplet and the particle [6], creating an irreversible adsorption between these two surfaces. This adsorption increases the stability of the emulsion [7], allowing for the development of different products using solid particles with adequate particle size and wettability properties by the immiscible liquids.

Different types of particles can create Pickering emulsions with diverse combinations of immiscible liquids, such as metallic particles [5], synthetic polymers [7], or natural biomacromolecules [8,9] such as cellulose. The use of cellulose as a Pickering emulsion stabilizer has received more attention in recent years as a potential surfactant replacement following the trend of younger consumers looking to buy food products perceived as having a greater sense of naturalness (branded as “clean label”) [10].

There is a growing interest in the research of Pickering emulsions. The number of published articles on Pickering emulsions and nanocellulose increased from 10 per year in 2014 to 105 in 2021 and 143 in 2022, showing an annual percentage growth rate of 17.99%. Additionally, almost 300 patents have been filled since the first one in 2010.

The objective of this review is to present a critical discussion about how research on cellulose, especially nanocellulose structures, is currently being carried out and the scientific and technological development of Pickering emulsions. Articles and patents were analyzed to determine the evolution of research trends worldwide. Recently published articles from a pool of publications in the literature were reviewed to showcase the type of scientific work performed regarding new/nascent and mature food technologies and products. Crucially, this paper also covers potential regulatory issues, the underlying phenomena regarding Pickering emulsions obtained with nanocellulose, and the possible environmental impacts of nanocellulose-based Pickering emulsions. 

## 2. Scientometrics Analysis

The search was performed in July of 2023 on the Scopus^®^ database. Documents with the terms “nanocellulose,” “Pickering,” and “emulsion” were the subject of our research. The date of publication was limited to between 1998 and 2025, the type of publication was limited to journals, and the language used was limited to English. Journals dedicated to review articles, such as “Comprehensive Reviews in Food Science and Food Safety”, “Critical Reviews in Food Science and Nutrition”, “Current Drug Discovery Technologies”, “Current Opinion in Colloid and Interface Science”, “Current Pharmaceutical Biotechnology”, “Advances in Colloid and Interface Science”, “Advances in Condensed Matter Physics”, “Advances in Materials Science and Engineering”, and “Advances in Physics”, were excluded. Subsequently, a bibliometric study was performed on the remaining articles, and a total of 655 articles were retrieved.

The bibliographic data were analyzed on Biblioshiny [11]. A total of 179 journals published articles dealing with the topic were considered. The number of authors was 18,912, with an average number of documents per author of 0.174. Among the countries corresponding to the authors, China stands out as the one with the largest number of articles (343), followed by Canada (70), the USA (62), and France (38) (Figure 1). The number of citations was led by China (6284), followed by France (3021), Canada (1467), and Finland (896). However, the average citation per document could be organized from highest to lowest in the following order: France (116.2), the Czech Republic (82.0), Finland (64.0), and the United Kingdom (45.1). Table 1 shows the top 10 journals. The journal with the highest number of articles was Carbohydrate Polymers (69), followed by Food Hydrocolloids (53).

The set of journals with publications on Pickering emulsions and cellulose mostly contain articles related to the use of cellulose or the physical–chemical principles governing these systems. The most predominant type of published research is scholarly research. There are also published applications, especially regarding food. Most of the articles describe the effects of different types of “cellulose” (827 articles) and “emulsification” (690 articles) on the “stability of the emulsions” (440 articles). Unfortunately, there is a lack of publications focusing on eco-friendly and economic circular approaches, as well as legal aspects. It is paramount to keep advancing research on fundamental science and applications, but it is also essential to keep conducted research that could help alleviate the environmental challenges within the field. This way, lawmakers can form an opinion to integrate new products into daily life.

Figure 1 is a net that depicts the worldwide collaboration with respect to the published articles. In the figure, the darker the blue hue, the higher the number of publications per country. Furthermore, the thicker the red lines, the higher the number of publications with coauthors from the connected countries, showing a higher interaction between countries in the research networks. There are five main clusters (East Asia, North America, South America, Europe, and Australia), and a great amount of interaction within these clusters can be observed. Nevertheless, the research between these clusters is mostly conducted between the leading countries of each group. There should be more overall interaction among the researchers of all of these nations because cellulose can be found worldwide in an enormous diversity of sources.

Africa and Latin America should begin developing research in this area to avoid solely exporting raw materials and importing manufactured goods. Researchers from countries in these areas have a high diversity of cellulose sources, and there should be funding for developing locally produced eco-friendly innovative goods. Developing products with a reduced need for foreign oil and other inputs may help the development of these regions via a circular bioeconomic approach. Developing countries and regions could use the results from previous research on nanocellulose for Pickering emulsion in basic and applied studies. Furthermore, scientists from these countries should be involved in future investigations into the effects of Pickering emulsions and their components on the environment and human health. 

## 3. Nanocellulose for Pickering Emulsions

Nanocellulose can be extracted from different biomass using different processes. Pre-treatment followed by bleaching is usually carried out to remove non-cellulosic components like lignin, hemicelluloses, sugars, proteins, and lipids from plant biomass before nanocellulose extraction [12]. One of the steps with the highest environmental impact in the production of cellulose nanomaterial is the bleaching step [13,14,15] because it uses chemicals and is an energy-intensive process. One way to overcome this drawback is by developing more efficient bleaching processes. Another way is to use types of biomass that do not demand harsher bleaching steps, such as tunicates or bacterial cellulose. Finally, another potential approach is not exhaustively bleaching the biomass. This approach aims to create lignocellulosic nanomaterials [8].

The chemical processes commonly used to extract nanocellulose include acid hydrolysis [12], deep eutectic solvent isolation [16], and enzymatic extraction [17]. These approaches usually produce cellulose nanocrystals (CNC) that are a few nanometers wide (5–20 nm) and a few hundred nanometers long (100–300 nm) [4,16]. The surfaces of CNC can be modified according to the isolation reagents used. For example, hydrolysis with sulfuric acid adds negative charges to the surface, improving the zeta potential of the nanocrystals. Nevertheless, hydrochloric acid hydrolysis breaks the cellulose without changing the crystalline surface [18]. The yield and characteristics of the extracted crystals also depend on the experimental protocol.

Physical or combined physical and chemical approaches are performed using different methodologies, such as disk grinding [19], microfluidization [20], and TEMPO ((2,2,6,6-Tetramethylpiperidin-1-yl)oxyl) oxidation [21]. Physically extracted cellulose usually takes the form of a 25–70 nm wide (but thousands of nanometers long) web or mat of long nanofibrillated cellulose (CNF) and intertwined chains [19].

Cellulose presents advantages over other biomacromolecules for Pickering emulsion preparation. For example, starch is an isomer of cellulose that is much more hydrophilic than cellulose. Starch is wettable by polar liquids. Nevertheless, nonpolar fluids cannot wet this biopolymer unless starch is chemically modified or gelatinized [22]. Therefore, nanocellulose structures may not require mandatory surface modification, while starch nanostructures do. Chitin may modulate immune responses [23], and chitosan requires the deacetylation modification of chitin before it is suitable to stabilize Pickering emulsions [24]. These characteristics of both biopolymers are shortcomings in comparison to cellulose for the formation of Pickering emulsions. Proteins may be denatured by food processing conditions like temperature or pH [25], making them unsuitable as Pickering system stabilizers. CNC and CNF may be extracted at higher temperatures and acidic conditions [12,14], making them viable options for Pickering emulsions in food products that proteins may not resist as proper additives.

It is worth mentioning that cellulose extracted from agricultural byproducts and waste products contributes to the circularity of the bioeconomy. Materials that are considered to be economic or environmental liabilities may be upcycled as innovative ingredients [15]. The valorization of this biomass may have positive economic impacts on producers, recyclers, industries, and consumers.

### 3.1. Cellulose-Based Pickering Emulsions Stability

The inter- and intramolecular hydrogen bonds between hydroxyl groups and the nonpolar interactions between carbon atoms stabilize the nanocellulose crystalline structure. Even if this domain is more rigid and stable than a paracrystalline domain, there is still mobility for atomic interactions, such as those that include hydroxyls of carbons C2 and C6 [26], or the crystal itself can be twisted [27]. The flexibility of the crystalline domains is primordial in order to stabilize Pickering emulsions.

One of the believed mechanisms for the stabilization of Pickering emulsions is that the nonpolar plane (2 0 0) from the crystalline regions bends over the droplet of nonpolar liquid, exposing hydroxyls to create hydrogen bonds with water molecules around the droplet [4]. If the hydrophobicity increases in the nanocellulose, water-in-oil emulsions are formed, and if the hydrophilicity increases, oil-in-water systems are formed [4]. Considering that there is a bending on the nonpolar plane following this model, shorter nanocrystals may have a better emulsifying capacity than longer ones. Indeed, shorter CNCs have been shown to result in smaller droplets [28] and have a better efficiency in stabilizing Pickering emulsions because shorter nanostructures have a higher packaging density and cover a larger surface area of the droplets, increasing steric obstruction between the droplets and resulting in a higher number of smaller oily structures [4]. A different mechanism seems to take precedence with CNFs. Since they are much longer than CNCs, the same nanofibril may concomitantly stabilize adjacent droplets of other nanofibrils [29], forming a network of nanofibril-stabilized droplets (Figure 2).

Charges also play a role in stabilization. Sulfuric acid may create negatively charged sulfur esters on crystals, impacting the zeta potential and resulting in more stable suspensions [18]. TEMPO oxidation converts the carbon-6 of cellulose into carboxylic groups [21], adding surface charges to the structure, increasing the repulsion between droplets, and improving the emulsion stability.

Different types of allomorphs may also affect the stabilization efficiency. Researchers have tested CNCs made of cellulose I (parallel chains, CNC-I) and II (antiparallel chains, CNC-II). In one study, although CNC-II was shorter and theoretically had a larger specific surface area, the droplets from CNC-I were smaller and required a lower content of nanostructures. Different crystalline structures reduced the hydrophobicity to CNC-II, resulting in less stable emulsions [30].

Cellulose nanostructures from different types of biomass can stabilize Pickering emulsions [9,20] with more versatility than other natural polymers. Nanocellulose can also be produced from partially delignified biomass, reducing energy costs compared to other biopolymers. Acid-extracted CNCs from pineapple peels with different contents of residual lignin were used to prepare oil-in-water Pickering emulsions with commercial sunflower oil. After 15 days of storage at 4 °C, the systems with 11% lignin on the CNCs presented smaller droplets than the emulsions prepared with 5% lignin on the CNCs. The emulsion prepared with the lower lignin content also had a water layer at the bottom of the flasks, indicating less emulsion stability [31]. Lignocellulosic nanofibrils were extracted with p-toluenesulfonic acid hydrolysis in two different temperatures, followed by disk grinder mechanical fibrillation from eucalyptus pulp, resulting in lignocellulosic nanofibrils with high (17%) and low (11%) lignin contents. The high-content lignin nanostructures created styrene droplets with smaller sizes, and the emulsion had a higher stability [8].

The process of nanocellulose extraction impacts the efficiency of the Pickering emulsion stability. Researchers and engineers should evaluate the most suitable options for their planned products by considering economic, environmental, and operational aspects. For example, using CNC from rice husks may be a fitting option in Asia, while CNF from birch may be an affordable choice in Europe.

### 3.2. Impact of the Phase Composition on the Stability of Pickering Emulsions

In another study, different types of oil (corn oil, sunflower oil, flaxseed oil, orange oil, Miglyol^®^ 812N, and fish oil) were tested under the same processing conditions to prepare oil-in-water emulsions stabilized by CNC extracted from wood by sulfuric hydrolysis. Orange oil had the smallest density, 0.84 g/cm^3^, while the density of the other oils ranged from 0.92 to 0.95 g/cm^3^. The emulsion with orange oil presented the largest droplet size. The droplet size of the emulsion prepared with flaxseed oil was statistically (*p* < 0.05) smaller than the other emulsions, even though flaxseed oil had characteristics that were derivative of triglyceride composition. The authors hypothesized that minor components of the flaxseed oil could be the reason for this difference, but this should be researched more extensively [32].

The ionic strength of the polar liquid may also impact the emulsifying properties. If the charge density of the cellulose structures is too high, the particles may lose the amphiphilic behavior and not form stable droplets [33]. One way to prevent this shortcoming is by increasing the ionic strength of the polar liquid by, for example, adding salts. The presence of 50 mmol/L of NaCl in water helped to stabilize the formation of a Pickering emulsion with CNC from MCC (microcrystalline cellulose) extracted by sulfuric hydrolysis. The salt disrupted the charge density of the crystals and eased their coverage on the oil droplets [30]. However, if the ionic strength of the polar phase is too high, the droplets may become unstable. The droplet size of TEMPO-oxidized bacterial cellulose could be tuned based on the pH of the continuous phase, changing the amount of protonation of the carboxyl groups or the addition of NaCl. The authors of [34] observed that the droplet size increased with growing salt concentrations from 25 mmol/L to 100 mmol/L, and NaCl concentrations higher than 100 mmol/L disrupted the network of TEMPO-oxidized CNF from oil palm empty fruit bunch produced with a microfluidizer, resulting in the coalescence of the droplets and creaming [35].

Although many experiments occur with distilled water, it is unlikely that a food emulsion will have the polar phase with no salt or other compounds dissolved in it. Food products may be acidic (e.g., orange juice) or have high ionic strength (e.g., sports drinks). It is essential to use a diverse range of polar media when studying the stability of the Pickering systems for food products. Additionally, researchers must pay closer attention to the other components and characteristics of the systems, such as proteins, surfactants, or other carbohydrates in the emulsifying properties of nanocellulose. These variables, which were not usually accounted for in most of the experiments we read about, may alter the interaction between the nanostructures and the phases. For example, changes in the pH may alter the surface charge, inducing the coalescence of droplets [33].

## 4. Applying Pickering Emulsions to Food

### 4.1. Foodstuffs

One of the main applications of Pickering emulsions is the production of emulsion-based food products. There are many articles (more than 650 documents) that relate to biopolymers (Section 2), including cellulose nanostructures, for this application. There are also patents regarding equipment and procedures that could be used to prepare Pickering emulsions for food applications [36,37]. In a recent search on the Google Patents website (6 September 2023), almost 280 patents were found with the terms “nanocellulose” and “Pickering”.

Pickering emulsions can be an intermediate step for preparing powdered oil, a material similar to powdered milk. In one study, a dispersible powder of oil-in-water camellia oil Pickering emulsion prepared with CNC from microcrystalline cellulose, extracted with a high pressure homogenizer using hydroxypropylmethylcellulose (HPMC) and sodium carboxymethyl cellulose (Na-CMC) was prepared via spray-drying. The Pickering emulsion was successfully dried, encapsulating the oily phase, and then redispersed in water, reforming the camelia oil emulsion [38]. These results indicated that HPMC and Na-CMC improved the stability of the emulsion prepared with CNC from microcrystalline cellulose. In another study, CNC from peach palm agricultural waste emulsified avocado oil. The authors compared the efficiency of using nanocellulose (1% *w*/*w*) with the efficiency of sorbitan monostearate (3.5% *w*/*w*). The stability indices (percentage ratio of the volume of the emulsion layer and total volume) after 72 h of the systems prepared with sorbitan were between 2.99 and 4.37 and the stability indices of the Pickering emulsions were always above 97% in all tested conditions for pH (3, 7, 11) and temperature (2 °C or 25 °C) [39]. The authors of this study point out that the stability of an emulsion in a broad range of pH values and temperatures is imperative when producing a food product. For example, a sauce may contain acidic additives such as organic acids or citric juices and may be stored at room temperature (≈25 °C) or refrigerated (≈4 °C). In the case of this product, a cellulose-based Pickering emulsion may be more efficient than a system prepared with traditional surfactants.

An exciting aspect of Pickering emulsions with respect to their potential food applications is that they could provide an approach to slow down the digestion of lipids while imbuing the desirable effects of those lipids into a food’s texture [40]. Pickering-stabilizing particles must resist gastrointestinal conditions (e.g., low pH in the stomach and the action of enzymes) to reduce lipid release to the intestine [41]. Cellulose is non-digestible by the enzymes in the upper part of the human gastrointestinal tract, an acidic environment with proteolytic enzymes. Therefore, nanocellulose materials appear to be promising structures in the context of stabilizing emulsions. In one study, a Pickering oil-in-water emulsion prepared with wood CNC reduced simulated corn oil small intestine digestibility from ≈60% to ≈20% [41]. This process maintains the same texture as regular emulsion-based products but can provide lower caloric intakes and the potential to reduce obesity.

Another application for Pickering emulsions is to improve the effective release and intestinal absorption of liposoluble bioactive food components. In one study, curcumin was dissolved in coconut oil. The oil droplets of this mixture were stabilized with CNC, forming an oil-in-water Pickering emulsion. Unfortunately, the Pickering emulsion prepared with curcumin, coconut oil, and CNC had a release percentage smaller than 20% after 24 h in the simulated stomach and intestinal fluids [42]. Alternatively, new research studies that make use of other cellulose nanostructures (CNCs or CNFs) or different oily phases may help improve curcumin release.

Another point of discussion is the ethical dilemma of preventing oil absorption in some foodstuffs while there is a shortage of calories for people in many countries. There is no easy or definitive answer to this question, but researchers and producers must keep this in mind to prevent future backlash to this technology. There are already concerns about the food safety of nanostructures in foodstuffs [1,3], and the best way to deal with misconceptions, prejudices, and fear is by releasing trustworthy information with critical thinking. This can be achieved only with more public and private funding and the extensive divulgation of scientific production. This information may also impact the GRAS (Generally Recognized As Safe) definition of the material, and this topic will be discussed later in this review.

Finally, from a more positive point of view, we reviewed articles about cellulose structures on foodstuffs and packaging, indicating that a plant-based emulsifier like nanocellulose have adequate economic costs because plant-based ingredients are usually cheaper than animal-based ingredients [1]. There might be favorable social costs if food is produced with less expensive ingredients than traditional emulsifiers, and it is possible to minimize the use of cold chains and processing costs, allowing for the production of low-cost and nutritional food locally. Finally, there may be environmental benefits, which will be discussed later in this document.

### 4.2. Food Packaging

Packaging technology is a fundamental part of food engineering. If proper packaging can better preserve food and reduce spoilage, there may be economic, social, and environmental advantages. Some articles have presented exciting results on how Pickering emulsions can be used to prepare or functionalize packaging materials such as films or foams with improved mechanical, thermal, and barrier characteristics [43,44,45,46,47,48,49]. An example of a packaging preparation in the literature is an oil-in-water Pickering emulsion consisting of styrene and nanocellulose in water that was polymerized into a composite film. This PS/CNF composite film presented higher tenacity, better thermal properties, and good optical transmittance compared to neat polystyrene [2]. Although the research on this subject is exciting, more scientists should work on this topic. Only 31 articles from the database of 655 retrieved articles discuss the potential applications of Pickering emulsions for food packaging.

Mixing polymeric matrices with cellulose nanostructures through a Pickering approach has advantages over simply blending the chemicals. During the preparation of the emulsions, the cellulose interacts better with the polymer, avoiding larger clusters of cellulose within the matrix [43]. A more uniform material is obtained when the liquid dispersion is cast and dried. In one study, three different approaches were used to prepare PLA/nanocellulose composite films: solvent casting, melting mixing, and Pickering emulsion. The best dispersion and interaction matrix/filler of the nanocrystals in the PLA matrix was observed using the Pickering approach. An oil-in-water suspension of PLA dissolved in CH_2_Cl_2_ and water was stabilized with wood CNC, resulting in a PLA film with improved mechanical properties [43].

Functional packages can also be prepared using the improved molecular interaction offered by Pickering emulsions. For example, in one study, LA/lignin films produced using a Pickering approach that involved oil-in-water suspensions of PLA/CH_2_Cl_2_ and lignin nanoparticle suspension had better dispersion than films derived from the melting mixing approach. The PLA/lignin films had good UV light absorption because of the aromatic rings in the lignin particles. The authors of this study suggest that these films can be used for the packaging of UV-sensitive products [44]. The authors of another study developed systems for the controlled release of natural antibacterial essential oils to improve the shelf-life of food products. These systems are based on cellulose nanocrystals and microfibrillated cellulose to encapsulate limonene, cinnamaldehyde, and eugenol, and all of the systems had an encapsulation efficiency of around 80% with 0.7% or more of nanostructured cellulose [45].

Freezing and cold chain are energy-intensive processes in food storage and preservation. Isobaric freezing is currently the dominant process, while isochoric freezing is a potential solution to reduce energy costs across the cold chain [46]. Thermal insulation can help to save energy in frozen or refrigerated food storage by reducing heat transfer and the need for refrigeration [46]. An oil-in-water TEMPO-oxidized wood nanocellulose suspension was used to create a Pickering emulsion paper (PCM paper) with entrapped paraffin droplets for heat management [47]. The PCM paper had a total enthalpy of 139 J/g between 15 °C and 35 °C, close to the value for pure paraffin (183 J/g). Heating and cooling down cycles indicated that the PCM paper survived for 100 cycles without paraffin leakage. A mass of 2 g of PCM paper was applied to a surface of 25 cm^2^ and heated with a simulated sun with an intensity of 1000 W/m^2^. The temperature under the surface with no covering changed from 25 °C to 35 °C in 5 min, while covering the surface with the PCM paper increased the time to 23 min [47].

In another study, cationized CNF improved the Pickering emulsion of methyl-methacrylate. The polymerized droplets resulted in a white latex-type emulsion. These microcapsules act as a phase change material for thermal energy storage because the PMMA absorbs heat and melts in warming conditions. In cooling conditions, the PMMA releases the heat and recrystallizes [48]. The future development of biodegradable packages with thermal functionalities may benefit a green and circular economy, increasing global energy efficiency. In another study, Pickering emulsions successfully formed foams [49] that may be used for thermal insulation and protection against mechanical impacts, helping to avoid food waste while reducing the environmental impact regarding keeping a food cold chain.

### 4.3. Innovative Food Applications

Plant-based foods are getting more attention worldwide for many reasons, including their potential to reduce environmental impacts [50] and animal welfare concerns [51] and avoid future zoonotic diseases [52]. Fungi-based and cell-based food may also replace animal-based food with additional environmental benefits [53,54]. In one study, a foam prepared with a Pickering emulsion was a suitable substrate for the cultivation of human neuroblastoma cells [49], and biodegradable foams may be a scaffold for cell- or fungi-based meat analogs with reduced environmental footprints. A Pickering emulsion stabilized with CNC was used as 3D printing ink in [55] to prepare custom-made structures. The authors of this study created oil-in-water Pickering emulsions with 80% oil that were stabilized with 0.5 wt% of CNC and suitable for 3D printing even with an ionic strength of 50 mM NaCl. The SEM of the printed materials is a scaffold of apparent high surface area [55]. This 3D-printed sponge may potentially be applied to grow cells and develop novel 3D food products.

Pickering emulsions based on cellulose may be used for many food applications. Nevertheless, these applications must have environmental benefits (e.g., biodegradability or lower footprints) to improve sustainability. Researchers must work to solve the current issues related to food production, storage, and consumption while always keeping regulatory and environmental issues in mind.

## 5. The Sustainability of Pickering Emulsions

Governmental and financial institutions worldwide have recognized that designing and developing sustainable technological solutions for the bio-based and circular economy is of essential importance [56]. It is of fundamental importance to achieve a successful sustainable innovation to alleviate safety, environmental, and socioeconomic issues at the early steps of conceptualization and experimentation of the processes and products.

Easier and cheaper modifications in techniques and material sources implemented during the early stages of product research, development, and innovation (RDI) may better reduce the environmental impacts of technological innovations [57]. However, at these stages, many unknowns are present in the technology development process, with uncertainties related to product properties and applications as well as processing equipment for mass production at an industrial scale being prevalent.

The consideration of safety, environmental, and socioeconomic issues require the development and adoption of a framework that describes the steps needed to identify hotspots and optimize the sustainability performance of novel products during the early research and development stages. In alignment with this need, the Joint Research Centre (JRC) of the European Commission recently proposed a framework to assess the safety and sustainability of chemicals and materials at the early development stage [58]. The aim was to prepare the chemical sector to attend the forthcoming European regulation for eco-design (the proposed Ecodesign for Sustainable Products Regulation—ESPR). This new regulation would repeal the current Ecodesign Directive 2009/125/EC (focused on energy-related products), which encompasses many types of products except those related to the food, feed, and medicinal sectors. This novel directive intends to support RDI teams to develop long-lasting, safer, and sustainable products for consumers and workers [56].

The Safe and Sustainable by Design (SSbD) framework [58] encompasses five steps that, when followed, help to implement safety and sustainability assessments in early maturity levels of chemical or material development (technology maturity level—TRL), including conceptualization (TRLs 1–2), experimentation (TRLs 3–5), pilot (TRLs 6–7), and industrial (TRLs 8–9). The three initial steps focus on the identification of potential chemical hazards (human health, environmental, and physical), while the last two steps involve analyzing environmental and social economic issues:The chemical intrinsic properties (step 1);Occupational health and safety in the workplace where the chemical/material is produced (step 2);Occupational health and safety at the site where the chemical/material is used (formulation of another product, step 3).Life cycle environmental impact assessment of the novel chemical (step 4);Socioeconomic impacts of the novel chemical (step 5).

Although the SSbD framework was originally designed to support the sustainable development and use of chemicals, it also applies to the sustainable development of other materials, such as nanocellulose Pickering emulsions, enabling the identification of product hotspots. In this way, this section explores the main issues that arise when applying this framework to the eco-design of nanocellulose Pickering emulsions (Figure 3).

### 5.1. The Occupational Health and Safety of Nanocellulose Pickering Emulsions

Reviewing toxicological studies concerning nanocellulose Pickering emulsions showed that these emulsions are generally considered non-toxic or only slightly toxic [1,3,59] for use in food products. Furthermore, an in vivo and in vitro analysis of nanocellulose Pickering emulsions showed that they can effectively deliver bioactive compounds and reduce lipid uptake in food applications, meaning that they have the potential to contribute to healthy diets. In one study, a CNC-stabilized Pickering emulsion successfully encapsulated functional ingredients (coumarin and curcumin) that presented anticancer and antimicrobial effects in cytotoxicity in vitro tests [41]. Simulation and in vivo tests have indicated that ingested nanocellulose with high-fat cream can reduce triglyceride hydrolysis and fat absorption, demonstrating the potential use of nanocellulose in diets that focus on weight loss [60].

Focusing on nanocellulose, in one study, small quantities of ingested CNF and CNC added to food caused little acute toxicity [61]. Furthermore, ingested CNF had few effects on the fecal metabolome and alterations in epithelial cell junction genes in a short-term experiment conducted using a rat gavage model [62].

Regarding the possible hazards for workers when producing and using nanocellulose Pickering emulsions in food companies (steps 2 and 3 in the SSbD framework), the major source of concern is nanocellulose intake through inhalation [63]. The few studies evaluating the effect of dermal and eye contact with nanocellulose (CNC and CNF) showed no significant result for all tested concentrations in in vitro cytotoxicity tests [64,65].

Initially, the Occupational Exposure Limits Value (OEL) for nanocellulose structures via inhalation or dermal exposure was recommended to be 0.01 fiber cm^−3^ (the same as for carbon nanofibers) [66]. This limit was proposed after considering the limited data availability, the potential biopersistence of nanomaterials when inhaled, and the precautionary principle.

Later, other studies compared the toxicity impacts of nanocellulose with other materials, presenting different results for oropharyngeal exposure [67,68]. In a study in which mice were exposed to gel and powder CNC and crocidolite asbestos, the authors concluded that the inhalation of nanocellulose structures caused dose-dependent oxidative stress, tissue damage, and robust lung inflammation. In comparison with asbestos, CNC causes more severe effects [68]. Other studies have investigated mice lung responses (inflammation and immune response) to four fibrous materials aspiration (CNC, CNF, carbon nanotubes, and crocidolite asbestos); one study found that nanocellulose caused discrete inflammatory mice lung response 14 days post-exposure, lower than asbestos and carbon nanotubes [67]. These two studies showed the influence of nanocellulose morphology and size on results and their dose–response effect on the lungs. Recent reviews of nanocellulose use in the food sector have highlighted the need for long-term exposure studies that feature actual workplace nanocellulose concentrations because few in vivo biological tests have been performed so far [69].

Although nanocellulose Pickering emulsions are generally considered to have low toxicity, the need for more studies to gain greater insights into their hazards and safe exposure limits has been highlighted in the literature [1,3,59]. The common argument is that nanocellulose may behave differently in the human body, causing biological effects that depend on particle physical–structural–chemical properties such as size, shape, charge, aggregation state, food matrix, and concentration [3].

As described in previous sections, many physical, chemical, and enzymatic processes are available to extract nanocellulose, mostly at lab or pilot scales, with each extraction resulting in nanoparticles with different morphologies and charges. These differences in nanocellulose structures affect the emulsion’s stability and its toxicological profile, and extensive biological tests are required to determine the potential hazards of emulsions and safe exposure levels for workers, final consumers, and ecosystems.

The diversity of the types of cellulose nanoparticles has delayed the development of safety profiles that list their potential hazards and characteristics; in other words, this delay is attributable to the plethora of different widths, lengths, surface charges, and residual lignin and hemicellulose contents that these nanoparticles can have [69]. In this way, more investigations are necessary to identify the effects of different types of nanocellulose on immunotoxicity, oxidative stress, genotoxicity, and nutrient uptake [70,71]. However, toxicological studies on food formulated with nanocellulose, such as Pickering emulsions, are as fundamental as the ones focused on nanocellulose.

Although cellulose can be modified in different harmless products like microcrystalline cellulose and hydroxypropyl methylcellulose, we would like to highlight some of the chemical reactions that can generate harmful byproducts that must be cleaned up after the reaction is concluded, such as lignosulfonates or strong acid wastes. Modified cellulose, such as nanocellulose, must be legally accepted as a GRAS (Generally Recognized as Safe) product by the official national list prior to being used in food emulsions. The United States Food & Drug Administration has published a valuable GRAS list, the Food Additive Status List [72].

In order for a new chemical to be considered GRAS, an organization has to ask the FDA to accept a new chemical as GRAS and provide enough evidence for its safe certification. A committee in the FDA will review and evaluate the presented documents before deciding to include a new chemical in the GRAS list. For example, ethyl cellulose has been considered GRAS since 2013, when the FDA analyzed the claim from Dow Wolff Cellulosics [72]. Even in this case, ethyl cellulose is approved as a food additive in only three situations. No organization have requested the FDA to analyze hydroxypropyl cellulose as GRAS, and this chemical is not listed as GRAS [73].

The most recent request to consider a nanocellulose derivative as GRAS was made for fibrillated cellulose in 2020 by Vireo Advisors, LLC (Boston, MA, USA), as the agent to the Alliance for Food Safety Acceptance of Fibrillated and Crystalline Celluloses (Alliance) on behalf of Borregaard AS (Sarpsborg, Norway), Evergreen Packaging, LLC (Lake Forest, CA, USA), Fiberlean Technologies Limited (Par, UK), Sappi Papier Holding GmbH (Hong Kong, China), Sappi North America Inc. (Boston, MA, USA), Sappi Southern Africa Limited (Johannesburg, South Africa), Stora Enso Oyj (Helsinki, Finland), and Weidmann Fiber Technology (Rapperswil-Jona, Switzerland) by Weidmann Electrical Technology AG (Dresden, Germany). The FDA did not consider the nanocellulose derivative to be GRAS in 2021 because of a lack of relevant information [73].

Considering the high diversity of nanocellulose types (CNC and CNF) from different sources currently available and the numerous vegetable oil types for formulating nanocellulose Pickering emulsions, it is essential to identify the most stable solutions. Then, for the selected nanocellulose materials, it is necessary to develop safety profiles and complementary toxicological studies (especially for human inhalation and dermal contact as this is particularly relevant for workers). These studies must follow the protocols used by major international governmental institutions, such as the European Chemical Agency (ECHA) and the FDA, to facilitate the inclusion of selected nanocellulose in chemical hazard categories (e.g., most hazard substances, substances of concern, other hazard properties; food GRAS). Moreover, toxicological tests for Pickering emulsions with selected nanocellulose need to be prioritized, considering the ingestion route for animal uptake.

### 5.2. Environmental Impact Assessment at Early Emulsion Development Stages

Regarding the environmental assessment of chemicals (Step 4 in the SSbD European framework), the environment life cycle assessment (e-LCA) method, which follows the framework of ISO 14040 and 14044, considers the whole product value chain from the extraction of raw materials up to product waste management (scope of cradle-to-end of value life or cradle-to-grave) [74,75]. Therefore, the function of the used material must be determined as early as possible in the RDI process.

Regarding the functionality of nanocellulose Pickering emulsions, Pickering emulsions could be used in the food sector to protect and continually release an active compound, such as carotenoids, in a juice. Waste management options for this emulsion may include collection and remanufacturing at the emulsion production company, composting, or digestion for biogas production. e-LCA results are always relative to the quantification of a product function in a functional unit. In the former emulsion example, the defined function unit could be the release of 3 g of carotenoids per 300 mL bottle of juice.

Thus, results change according to the defined function and function unit, and analyzing the most probable functions and functional units is required when assessing the impacts of novel products. Defining product functionality at the early product experimental stage, as recommended by the European SSbD framework [58], can be challenging because the focus of a research team is initially placed on a product’s proof of concept and defining its best processing route, meaning that little is known regarding its final application.

Although it is fundamental to consider the whole product value chain, a research team may start step 4 with a cradle-to-gate assessment to select the best performing alternative among production routes and/or materials [15,76]. The cradle-to-gate scope focuses on raw material extraction and the production phase, disregarding use and end-of-value life. This assessment scope usually uses a mass function unit, such as the production of 1 kg of a novel product.

For the food emulsion example, at the early experimental stage (TRL 3–4), the environmental impacts of alternative processing routes, types of nanocellulose (e.g.,: CNC or CNF from different plant sources), vegetable oils (e.g.,: sunflower or soy), or functional adjuvants may be compared to produce 1 kg of emulsion (stabilized for a predefined time). Such a comparison would allow for the selection of the best performing emulsion formulations. Later, at the end of the experimentation stage (TRL 5), the product use and end-of-life phases are more easily defined and evaluated. At these stages, the relevant processes have already been optimized, and more information regarding the product’s properties is available, allowing for the research team to focus on decisions regarding the product’s function and final destination.

The e-LCA of novel products performed at the early product development stages (ex-ante or prospective LCA) also requires special care when collecting inventory data and comparing environmental impact results with similar incumbent products. At these early stages, the quantification of inputs and outputs, when available, is at a lab scale. Collecting lab inventory data is essential for modeling production at the pilot scale, but data at this scale should not be directly used to assess environmental impacts since machinery and yield usually change with process upscaling [76]. The identification of critical points in a product value chain and the comparison of impact results between novel and incumbent products may be misleading at the lab scale, requiring process modeling at the pilot scale and scenario analysis [76,77].

The application of scenario analysis should be widely common for ex-ante LCA studies for the foreground (novel processes under development) and background (processes related to the rest of the product value chain, e.g., energy production) processes. A range of alternative scenarios can be defined for the foreground processes to compare future upscaled processing routes, product functionalities, end-of-life options, and the environmental impacts of novel and similar incumbent products [78]. Moreover, a range of future scenarios for the background processes should be integrated from Integrated Assessment Models (IAMs), and this could possibly involve applying a superstructure approach [79].

For assessing the environmental impacts of a novel product, a minimum of sixteen categories are proposed in the SSbD framework [58]. These categories encompass toxicity (human and freshwater ecotoxicity), pollution (particulate matter, ionizing radiation, ozone depletion and formation, eutrophication—terrestrial, marine, and freshwater, and acidification), and resource depletion (metals, minerals, fossil, land, and water). The methods outlined for assessing these environmental impact categories are the same as those outlined for assessing the product’s environmental footprint (PEF) [80].

It is important to mention that the toxicity impacts evaluated at this step are distinguished from those at steps 1–3. At this step, toxicity relates to all emissions, leaving production processes in a value chain, potentially affecting human health and ecosystems via multi-environmental compartments (e.g., soil, water, air) contamination. On the other hand, steps 1–3 evaluate the potential hazards that may arise when workers and end-users are directly exposed to the chemicals/materials.

### 5.3. Environmental Impacts of Emulsions and Main Components

So far, no environmental assessment of nanocellulose Pickering emulsions has been performed. Nonetheless, during the last ten years, some e-LCAs of Pickering emulsion components, including CNC [13,15,81,82,83], CNF [14,57,84,85,86,87,88], and vegetable oils [1,89,90], have been performed (Table 2).

Overall, the e-LCAs of nanocellulose adopted a cradle-to-gate scope, analyzing impacts for producing 1 kg of nanocellulose from different vegetable sources (wood, cotton, coconut husk, carrot waste, and sugarcane bagasse). In each study, it was worth comparing alternative materials and processing routes, allowing for research teams to indicate the best performing ones. However, the reviewed studies evaluated different impact categories, applying different methods and complicating the comparison and identification of the best performing materials.

Considering the inventories from Agrifootprint 5, we applied Recipe 2016 to evaluate the impact of different vegetable oils on climate change, observing that the values ranged from 0.79 (sunflower oil) to 10.45 kg CO_2_ eq/kg (palm kernel oil); thus, there was almost a ten-fold difference between them. Moreover, the effects of 1 kg of rapeseed (1.9 kg CO_2_ eq), rice bran (1.61), sunflower, and lupine (1.46) refined oils are smaller than that of 1 kg of CNF from carrot waste (2.9) at the pilot scale [84]. As the mass of vegetable oil is usually higher than that of nanocellulose in a Pickering emulsion, it is important to prioritize these oils, which have a low impact on the climate, when designing novel emulsions.

From this discussion about environmental ex-ante LCA, the need for future studies focusing on nanocellulose Pickering emulsions is clear. We encourage researchers to assess the proof of concept of products using a cradle-to-gate approach from the product prototyping stage or conduct a cradle-to-end-of-life assessment. Moreover, evaluating multiple scenarios for functional units and processing techniques using the most frequently used impact methods, such as ILCD/PEF (International Life Cycle Data System/Product Environmental Footprint), is indispensable.

### 5.4. Socioeconomic Assessment

The last step (5) in the SSbD framework is the socioeconomic impact assessment of chemicals and materials [58]. For social LCA (s-LCA), the Life Cycle Initiative from the United Nations Environment Programme (UNEP) presented a consensus method for assessing the potential life cycle social impacts of products based on the same four iterative phases described in ISO 14040, which is largely used to perform e-LCA [95]. This method presents a hierarchical structure connecting five stakeholders to six impact categories, forty subcategories, and alternative indicators for each subcategory. Moreover, other similar protocols have been proposed by the World Business Council for Sustainable Development [96] for chemical products and by the Roundtable for Product Social Metrics [97] for general products.

The s-LCA requires evaluating aspects that affect stakeholders in the product value chain. Farmers, workers, consumers, and society are crucial stakeholders in the food sector. However, the main stakeholders that are considered in social assessment tools for the food sector include workers, local communities, and consumers [98]. Moreover, some s-LCA case studies have also only considered workers and local communities [95]. In this way, the SSbD framework prioritizes workers, local communities, and consumers and the associated impact subcategories for use in s-LCA studies [58]:Workers: child labor, fair salary, forced labor, health, and safety (focusing on accidents at the workplace since other aspects are considered in steps 2), freedom of association and collective bargaining, working hours, and equal opportunities/discrimination;Local community: community engagement and local employment;Consumers/users: health and safety (not covered in step 3) and accountable communication.

Few s-LCA studies have been performed for the food sector, and most s-LCA studies are related to incumbent products with well-established value chains [99,100]. Thus, food emulsions, emulsifiers, and nanocellulose have not been the subject of these studies. Due to the novelty of the proposed s-LCA methods, research teams need time to adapt them to the product development stages.

Regarding economic impact assessment, the SSbD framework involves applying environmental Life Cycle Costing (LCC) [58]. Conventional LCC calculates the total costs of a product life cycle according to the study scope, encompassing raw material acquisition, plant operation and maintenance, product use, and final disposal in cradle-to-end of value life studies. On the other hand, environmental LCC considers external environment costs related to the prevention (e.g., recycling or waste reduction) or promotion (e.g., resource depletion or pollution) of environmental impacts in the value chain. External environmental costs are calculated by monetizing environmental impacts, a challenging task for decision-making when different stakeholders are involved.

Unlike e-LCA and s-LCA, there is no standardized framework for performing LCC studies (conventional or external). A recent review of LCA studies that integrate environmental and economic impacts showed how they adopted conventional LCC, summing costs per functional unit considering the process inside the scope of each study [101]. Most studies were cradle-to-gate studies, comparing products, processes, or projects using different cost categories (e.g., direct and indirect, internal and external, and operational and non-operational costs).

No LCC studies on nanocellulose Pickering emulsions were found in the scientific literature. However, for nanocellulose, three of the studies we reviewed considered economic analysis when comparing alternative processing routes: a CNF study applying conventional LCC [84], a CNF study considering operational costs in an ecoefficiency analysis [92], and a CNC study using cash flow analysis [81].

As mentioned for e-LCA, performing an s-LCC and LCC of a novel product (such as Pickering emulsions) at an early experimental stage could necessitate the adoption of a cradle-to-gate scope since data related to emulsion final technical characteristics and use are not available yet. Conventional LCC may be applied at this stage and expanded to consider external costs later at the pilot stage.

Moreover, eco-design teams need to define multiple scenarios for performing LCAs. The formation of a team with knowledge in multiple disciplines is fundamental for developing relevant scenarios for decision-making. Such a team shall define important stakeholders, choose relevant impact categories, and estimate the future costs of materials, equipment, and end-of-life alternatives since industrial production and product consumption will happen years ahead of current research activities.

## 6. Conclusions and Recommendations

The bio-based circular economy is an essential element for the sustainable development and continuity of humankind. This review presented nanocellulose as a viable nanoparticle for preparing Pickering emulsions. The reviewed articles are evidence that this is a feasible food technology.

Unfortunately, there are unsolved issues in several fields. Cellulose-based Pickering research has, so far, only been conducted in a few countries, especially developed and emerging countries, with Asiatic and African nations comparatively making fewer contributions to the literature. Recent research and research trends have mainly focused on material science and engineering issues. Although this is important to create new products, there is a need for the development of scaling-up technologies that allow for the production of kilograms or liters per hour of products.

Moreover, research on the sustainability and eco-design of products with Pickering emulsions is needed. There are LCA studies regarding emulsion components but not of emulsions themselves. More interaction is needed among health, material sciences, and food engineering researchers to address the complex issues regarding approving a food containing Pickering emulsions. Ultimately, legal professionals must be involved in the discussion of new food products to prevent public backlash and government misunderstanding, especially when these products have reached the technological maturity required for commercialization.

Finally, consuming countries of this technology must approve a legal framework that allows food products with Pickering emulsions to be put on the market. Solving these shortcomings must happen concomitantly to allow for the successful release of these types of food products on the market. The partnership between private and governmental sectors will be of utmost importance to achieve these goals. If this system succeeds, it may provide a reference point for other innovative products, speeding up technology adoption and economic innovation and benefitting society and the environment.

## Figures and Tables

**Figure 1 foods-12-03599-f001:**
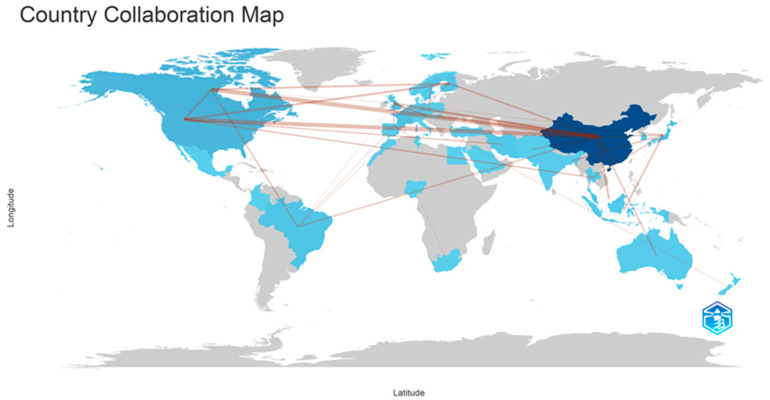
Map of interaction between countries based on the authors of the articles on “emulsion AND Pickering AND cellulose” after limiting our search to articles published in journals in English retrieved in July 2023 from the Scopus^®^ database. Figure generated with permission from the ref. [11] Open Access GPL-3. 2017. University of Naples.

**Figure 2 foods-12-03599-f002:**
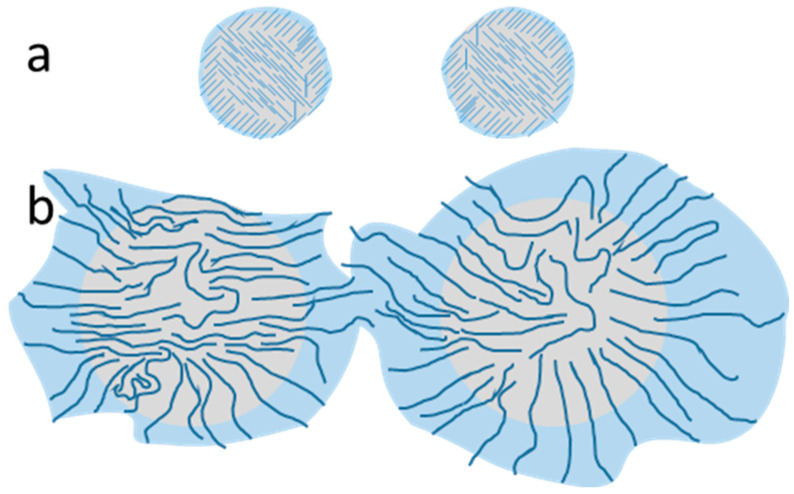
Difference between droplets coated with CNC and CNF. The oil droplets (gray spheres) may be coated by (**a**) CNC (straight dark blue lines) or (**b**) CNF (squiggly dark blue lines). The CNC has a higher coverage on the oil and creates smaller droplets, but the area of influence (light blue spot) from the rods is smaller, keeping each droplet as an individual entity. The CNF has a smaller coverage on the oil and creates larger droplets, but the area of influence from the nanofibers is larger, eventually merging with the area of influence from another droplet and creating a network of individual entities.

**Figure 3 foods-12-03599-f003:**
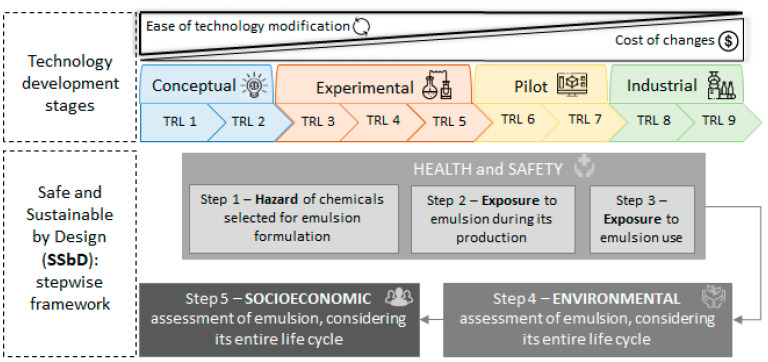
SSbD framework applied to nanocellulose emulsions. Adapted with permission from refs. [57,58]. Ref. [57]: Open Access. 20217. Wiley Online Library. Ref. [58]: CC BY 4.0. 2022. Publications Office of the European Union. TRL: Technology Readness Level.

**Table 1 foods-12-03599-t001:** Number of published articles on “emulsion AND pickering AND cellulose” after limiting our search to articles published in journals in English retrieved in July 2023 from the Scopus^®^ database.

Published Articles	Sources
69	Carbohydrate Polymers
53	Food Hydrocolloids
40	International Journal of Biological Macromolecules
32	Cellulose
28	Colloids and Surfaces A: Physicochemical and Engineering Aspects
24	Langmuir
21	Journal of Colloid and Interface Science
18	ACS Applied Materials and Interfaces
16	Biomacromolecules
14	Food Chemistry

**Table 2 foods-12-03599-t002:** e-LCA of different types of cellulose and oil suitable for preparing Pickering emulsions.

References *	Feedstock	Processing Routes	Scale	Impact Categories	Environmental Impact Assessment Methods	Best Results for GWP ** for 100 years (kg CO_2_-eq/kg of Nanocellulose or Oil)
**CNC**						
(Figueirêdo et al., 2012) [13]	Coconut fiber	EUC—Chopping, washing, beaching (NaClO_2_), hydrolysis (H_2_SO_4_), centrifugation, dialysis.	Lab	Global warming potential/climate change (100 years), water depletion (total blue water consumption), eutrophication, and human toxicity	Recipe	122 for EC
	Cotton linter	EC—Chopping, hydrolysis (H_2_SO_4_), centrifugation, dialysis.				
(Nascimento et al., 2016) [15]	Coconut fiber	CNH1: gridding, pulping, bleaching, hydrolysis with diluted acid, dialysis.	Lab	Global warming potential/climate change (100 years), acidification, freshwater eutrophication, marine eutrophication, human toxicity and water depletion (total blue water consumption).	Recipe	207 for CNU
		CNH2: gridding, pulping, bleaching, hydrolysis with concentrated acid, dialysis.				
		CNO: gridding, pulping, oxidation with ammonium persulfate.				
		CNU: gridding, pulping, bleaching, and high power ultrasound.				
(Leão et al., 2017) [82]	Sugarcane bagasse	I—NaOH/NaClO_2_/H_2_SO_4_/60 min/4x.	Lab	Global warming potential/climate change (100 years) and water footprint (total water consumption = blue+ green + gray quantities)	CML 2010	13.7
		II—NaOH/NaClO_2_/H_2_SO_4_/60 min/1x.				
		III—NaClO_2_/NaOH/H_2_SO_4_/30 min/4x.				
		IV—NaClO_2_/NaOH/H_2_SO_4_/30 min/4x.				
		V—NaClO_2_/NaOH/H_2_SO_4_/30 min/1x.				
		VI—NaOH/NaClO_2_/HNO_3_/H_2_SO_4_/60 min/4x.				
		VII—NaOH/NaClO_2_/HNO_3_/H_2_SO_4_/60 min/1x.				
		VIII—NaClO_2_/NaOH/HNO_3_/H_2_SO_4_/30 min/4x.				
		IX—NaClO_2_/NaOH/HNO_3_/H_2_SO_4_/30 min/1x.				
		X—NaOH/H_2_O_2_/H_2_SO_4_/30 min/4x.				
		XI—NaOH/H_2_O_2_/H_2_SO_4_/30 min/1x.				
		XII—NaOH/H_2_O_2_/H_2_SO_4_/30 min/1x.				
(Bondancia et al., 2022) [81]	Sugarcane bagasse	S-CNC: pretreatment (Hydrothermal enzymatic hydrolysis, organosolv, bleaching) and hydrolysis with sulfuric acid.	Modeled industrial scale	Global warming potential/climate change (100 years)	CML-IA Baseline v. 3.04 2000	No real values presented but indication of S-CNC as best case (twice lower impact than other routes).
		Cit-CNC: pretreatment (Hydrothermal enzymatic hydrolysis, organosolv, bleaching) and hydrolysis with citric acid.				
		Cit-S-CNC: pretreatment (Hydrothermal enzymatic hydrolysis, organosolv, bleaching) and hydrolysis with a combination of acids.				
(Zhang et al., 2022) [83]	Cotton	1—Sulfuric acid hydrolysis, followed by the separation of sulfuric acid from hydrolysate mixture using gravity settling, followed by microfiltration, tubular centrifugation, and ultrafiltration.	Pilot	Global warming potential/climate change (100 years) aquatic ecotoxicity, terrestrial ecotoxicity, aquatic acidification, aquatic eutrophication, and non-renewable energy.	IMPACT 2002+	30 for route 4
		2—Sulfuric acid hydrolysis, followed by the separation of sulfuric acid from hydrolysate mixture using gravity settling, followed by disk stack centrifugation, microfiltration, tubular centrifugation, and ultrafiltration.				
		3—Sulfuric acid hydrolysis, followed by the separation of sulfuric acid from hydrolysate mixture using low-speed centrifugation, followed by disk stack centrifugation, microfiltration, tubular centrifugation, and ultrafiltration.				
		4—Sulfuric acid hydrolysis, followed by the separation of sulfuric acid from hydrolysate mixture using ceramic membrane microfiltration, followed by disk stack centrifugation, microfiltration, tubular centrifugation, and ultrafiltration.				
**CNF**						
(Li et al., 2013) [86]	Delignified Kraft pulp from wood	TOSO: TEMPO-oxidation for chemical modification; sonication for mechanical disintegration.	Lab	Global warming potential/climate change (100 years), carcinogens human health, respiratory organics, respiratory inorganics, radiation, ozone layer, ecotoxicity ecosystem quality, acidification/eutrophication, land use, mineral resources, and fossil fuels.	Eco-Indicator 99	190 for TOHO route
		TOHO: TEMPO-oxidation for chemical modification; homogenization for mechanical disintegration.				
		CESO: chloroacetic acid etherification for chemical modification; sonication for mechanical disintegration.				
		CEHO: chloroacetic acid etherification for chemical modification; homogenization for mechanical disintegration.				
(Arvidsson et al., 2015) [91]	Wood pulp	1—Enzymatic pretreatment, followed by microfluidization.	Lab	Energy use, global warming potential, acidification, and water use.	CED for energy use and Recipe for the other categories	0.79 for route 1
		2—Carboxymethylation pretreatment, followed by microfluidization.				
		3—No pretreatment route, followed by homogenization.				
(Piccinno et al., 2015) [88]	carrot waste (by-product of carrot juice production)	Boiling, enzymatic depolymerization, and homogenization.	Lab	Global warming potential, fossil fuel depletion potential, freshwater ecotoxicity potential, human toxicity potential, ionizing radiation potential, marine ecotoxicity potential, ozone depletion potential, photochemical oxidant formation potential, terrestrial acidification.	ReCiPe	107
(Piccinno et al., 2018b) [92]			Pilot	Human health, ecosystem quality and resources	ReCiPe endpoint	32 ***
(Nadeem et al., 2022) [87]	Refining	1—Refining (milling), homogenization, and high-pressure homogenization.	Lab	Global warming potential/climate change (100 years), energy demand, and water depletion (total water consumption)	IPCC, CED and Water depletion	20 for route 1
		2—Refining (milling) and homogenization.				
		3—High-pressure homogenization.				
(Berglund et al., 2020) [84]	Carrot waste (by-product of carrot juice production)	Pretreatment (distilled heated water, alkali treatment with NaOH and bleaching with NaClO2), followed by gridding stone fibrillation (ultra-fine grinder and high-pressure homogenizer).	Pilot	Acidification, global warming potential/climate change (100 years), ecotoxicity freshwater, eutrophication freshwater, eutrophication marine, eutrophication terrestrial, human toxicity/cancer effects, human toxicity/non cancer effects, ionizing radiation/human health, land use, ozone depletion, particulate matter/respiratory inorganics, photochemical ozone formation, resource depletion water, and resource depletion/minerals and metals, and resource depletion/fossils, and resource depletion/renewables.	ILCD/PEF	2.9 (high-pressure homogenizer and European energy mix)
(Haroni et al., 2021) [14]	Sugarcane bagasse	Two different pretreatments for removing lignin (alkaline NaOH or organosolv) and two different methods for depolymerization (TEMPO oxidation or lime juice hydrolysis).	Lab	Global warming potential/climate change (100 years).	IPCC 2013 GWP 100a	4.45 (alkaline delignification and TEMPO oxidation)
(Krexner et al., 2022) [85]	Manure (maize silage and pig slurry)	Anaerobic fermentation of manure (biogas and cellulose-residue), pulping (Kraft bleaching), and grinding (high-pressure homogenizing).	Pilot (fermentation and pulping), lab for CNF extraction	Global warming potential/climate change (100 years), fossil resource scarcity, freshwater eutrophication, human toxicity, terrestrial acidification (TAP) and terrestrial ecotoxicity potential.	ReCiPe 2016	4.41
**Vegetable oil**						
(Bai et al., 2021) [93]	Soybean	Conventional irrigated crop production and oil refining in China.	Industrial	Global warming potential/climate change (100 years), ozone formation-human health, fine particulate matter formation, ozone formation, stratospheric ozone depletion, terrestrial acidification, freshwater eutrophication, marine eutrophication, freshwater ecotoxicity, marine ecotoxicity, human carcinogenic toxicity, land use, fossil resource scarcity, terrestrial ecotoxicity, human non-carcinogenic toxicity, ionizing radiation, water consumption.	ReCiPe 2016	5.67
	Rapeseed					3.35
	Peanut					3.15
(Prado et al., 2021) [90]	Cotton	Conventional crop production and oil refining in US.	Industrial	Global warming potential/climate change (100 years), abiotic depletion, eutrophication, acidification, photochemical oxidation, fine particulate matter, ozone layer depletion, water scarcity.	IPCC (2007) for climate change, AWARE (Water scarcity), Recipe 2016 for Fine particulate matter, and CML-IA for the other impact categories.	1.1
	Soybean	Conventional crop production and oil refining (global average).				6.4
	Canola	Conventional crop production and oil refining in US.				3.8
	Palm	Conventional crop production and oil refining in Indonesia and Malaysia.				4.8
(Fernández-Lobato et al., 2022) [89]	Olive oil	Olive cultivation systems in Tunisia (extensive, intensive, and super-intensive) as well as the main extraction systems (press, 3-phase and 2–3 phase-combined system).	Industrial	Acidification, global warming potential/climate change (100 years), ecotoxicity freshwater, eutrophication freshwater, eutrophication marine, eutrophication terrestrial, human toxicity/cancer effects, human toxicity/non cancer effects, ionizing radiation/human health, land use, ozone depletion, particulate matter/respiratory inorganics, photochemical ozone formation, resource depletion, water and resource depletion/minerals and metals, and resource depletion/fossils, and resource depletion/renewables.	ILCD/PEF	3.53 (most representative system: extensive crop production and refining with three phase extraction)
(van Paassen et al., 2019) [94]	Refined coconut oil		Industrial	Global warming potential/climate change (100 years).	Recipe 2016	3.33
	Refined rice bran oil					1.61
	Refined rapeseed oil					1.90
	Refined maize germ oil					0.86
	Refined palm oil					7.44
	Refined palm kernel oil					10.45
	Refined sunflower oil					0.79
	Refined soybean oil					5.13
	Refined lupine oil					1.46

* All references adopted a “cradle-to-gate” boundary. ** GWP: Global Warming Potential. *** The value was estimated considering at lab scale; 107 kg CO_2_ eq/kg was related to a final score of 6.7. At pilot scale, the final score was 2 for the 1C scenario [92].

## Data Availability

The raw/processed data required to reproduce the above findings cannot be shared at this time due to legal/ethical reasons.

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
