# Peer review of "Sustainable Pickering Emulsions with Nanocellulose: Innovations and Challenges"

_foods, 2023, doi:10.3390/foods12193599_

Round 1

Reviewer 1 Report

The work exposes the current work on Pickering emulsions stabilized by cellulose, their relevance, applications related to the food industry and the sustainability concerns and the research performed to provide information to fill the knowledge gap. 

The work is well-planed and interesting. It could be improved b including figures in the last sections, as they are a bit dense and a diagram, or other type of graphic could aid in explaining some of the concepts.

The results of the papers could also be used to provide a dimension to some of the claims and underscore their meaning and relevance. 

Introduction

This section shall be improved, it is important to highlight the importance of this work, preferably with data and numbers to support the observed trend. 

L31-32. I am unsure if the most accurate term is “Pickering micelle”, I think that it is more accurate to indicate that the stabilization mechanism for Pickering emulsions are fundamentally different to the phenomenon observed in traditional emulsions.

L36-39. The redaction of these lines is confusing. I think that it is best to indicate that the type of emulsion stabilized by solids depends on the contact angle of a single layer of particles in the water-oil interface, when the angle of these particles towards the water phase lies between 15 and 90 °, it stabilizes oil in water emulsions, and if it is between 90 and 165 °, it stabilizes water in oil emulsions.

Scientometrics analysis

L68. Please check for the parenthesis that appears on this line. I fail to see where it begins.

L102. Please indicate what is the meaning of the line that connect the different countries.

L110-111. Please connect these sections, they appear separated and the message that is trying to convey might get lost.

Nanocellulose for Pickering emulsions

L122-126. I think that it is important to mention in this paragraph that the surfaces of CNC can be modified according to the isolation reagents used, altering its physio-chemical behavior.

L124-125,130. Please include a size range to more accurately describe the morphology of the CNC and CNF.

L132-138. Please revise the clarity of this paragraph, as the writing in not clear enough.

L154. Please provide a reference.

L182-183. It is important to identify the criteria used to mention that the Pickering emulsifying stability was better, and include the data to support this affirmation.

L187-188. Please provide a value of the reduction observed in the droplet size of the lignocellulosic nanofibrils with high lignin, compared to the ones with a lower lignin content.

L189. I suggest to reconsider the name of this section. Perhaps, effect of the phase composition in the stability of Pickering emulsions or Impact of the phase composition in the stability of Pickering emulsions.

L190-192. It is unclear the type of oils evaluated, and the effect of other characteristics of the oil phase, such as chain size and polarity of the oil or mixture of oils. This section shall be improved with more in-depth information.

L209-L213. According to this paragraph, it would be valuable to mention if there is an effect of the pH and the presence of other components, such as proteins, surfactants or other carbohydrates in the emulsifying properties of nanocellulose.

Food application of Pickering emulsions

L216-220. Please indicate an approximate number of patents, to reinforce this paragraph.

L222-225. It is important to mention that HPMC and Na-CMC are stabilizers to improve the stability of the emulsion.

L228. Please include the range, and, if possible, the time the emulsion remained stable or the increase in this time observed in CNC and sorbitan monostereate.

L229. It is not clear which property is imperative.

L237. Please provide a reference of the conditions that are relevant during the processing through the gastrointestinal tract.

L240. Please provide the percentage of digestibility reduction.

L242. I am not sure if method is the most appropriate word here, maybe approach or application could be better in this situation.

L251-252. It is not custom to use the question mark in the paragraph. The ethical dilemma can be highlighted by other means.

269-272. While packaging and food packaging are a very active subject among the applications of nanocellulose, maybe, the use of Pickering emulsions as precursors might not be as active. It would be interesting to complement this paragraph with the results obtained in the Scientometric analysis.

L272. I think these lines shall be revised, as the comparison is not clear, and does not necessarily hold true, as it depends on which material is being compared and what is understood by “efficient”, is the efficiency related to the processing of the material or is it related to the barrier performance.

L281. Please provide a reference.

L301. Please indicate the main result obtained in that article, if possible, indicating the thermal conductivity or other measure of the thermal isolation provided by the packaging material.

L317-318. I am wondering if this application should be included in the section pertaining to packaging materials, also it would be interesting to briefly mention the results obtained with these packages.

Sustainability of Pickering emulsions

L361-362. Please maintain the format of the previous steps.

L393-400. There is a coherence issue with these lines. The authors indicate that oral evaluations have been conducted, however, they proceed to explain the effect of nanocellulose inhalation in lung cells.

L418. Please check this line, it is not clear.

L431. Please verify the redaction of this line.

L433. Please provide a reference. Also, it is important to keep in mind, that ethylcellulose is approved as a food additive in only three cases, as indicated by the FDA (https://www.accessdata.fda.gov/scripts/cdrh/cfdocs/cfcfr/CFRSearch.cfm?fr=172.868).

L542. Please indicate what does ILCD/PEF stands for.

Conclusions

The section about sustainability of Pickering emulsions could be mentioned more in this section, for example, the existence of the LCA of oils and cellulose is not mentioned.

It would be interesting to specify the type of research required to improve the calculation of the environmental and social impacts for the incorporation of Pickering emulsions involving nanocellulose, considering the intricate complexities of this subject.

The use of the language through the text could be improved, there are cases of awkward phrasing and, more importantly, some sentences are difficult to read, which reduces the comprehensibility of the message they are trying to convey. 

It is important to verify the language and make the changes accordingly.

Author Response

Review Report Form – Reviewer 1

Comments and Suggestions for Authors

The work exposes the current work on Pickering emulsions stabilized by cellulose, their relevance, applications related to the food industry and the sustainability concerns and the research performed to provide information to fill the knowledge gap. 

The work is well-planed and interesting. It could be improved b including figures in the last sections, as they are a bit dense and a diagram, or other type of graphic could aid in explaining some of the concepts.

The results of the papers could also be used to provide a dimension to some of the claims and underscore their meaning and relevance. 

Answer: Thanks for your feedback. We added a framework in the last sections to make the text easier to understand.

Introduction

This section shall be improved, it is important to highlight the importance of this work, preferably with data and numbers to support the observed trend. 

Answer: Thanks for your comment. We added in the introduction brief comments about the number of published articles and filled patents. Additionally, we showed data regarding the growing rate of article publications, as follows: “There is a growing interest on the research of Pickering emulsions. The number of published articles on Pickering emulsions and nanocellulose increased from 10 per year in 2014 to 105 in 2021 and 143 in 2022, showing an annual percentage growth rate of 17.99. Additionally, almost 300 patents were filled since the first one in 2010.”

L31-32. I am unsure if the most accurate term is “Pickering micelle”, I think that it is more accurate to indicate that the stabilization mechanism for Pickering emulsions are fundamentally different to the phenomenon observed in traditional emulsions.

Answer: Thanks for highlighting this point for improvement. We replaced “micelle” with “system”, kept the term “micelle” only for traditional emulsions, and added more information about the difference between the stabilization mechanisms: “Pickering emulsions are systems getting more awareness in the food field [1–3]. The stabilizing mechanism of a Pickering system is different from the mechanism commonly observed for micelles formed by immiscible liquids like oil and water [4]. Most conventional emulsions use some surfactant or tensoactive chemical as stabilizer, reducing the surface tension between the polar and nonpolar phases. Pickering emulsions, on the other hand, are formed by solid stabilizing particles with polar and nonpolar domains that act as a physical barrier between the phases, preventing the coalescence of droplets.”

L36-39. The redaction of these lines is confusing. I think that it is best to indicate that the type of emulsion stabilized by solids depends on the contact angle of a single layer of particles in the water-oil interface, when the angle of these particles towards the water phase lies between 15 and 90 °, it stabilizes oil in water emulsions, and if it is between 90 and 165 °, it stabilizes water in oil emulsions.

Answer: Thanks for the advice. We paraphrased your suggestion and rewrote the sentence, making it clearer: “A chemical coats the interface between the droplet (dispersed phase) and the dispersing medium (continuous phase), with no solubilization in any of the phases [5], usually forming a single layer of solid particles. When the contact angle of one of these single particles towards the water phase lies between 15° and 90°, an oil-in-water Pickering emulsion is formed, and if the angle ranges from 90° to 165°, it stabilizes a water-in-oil emulsion [6].”

Scientometrics analysis

L68. Please check for the parenthesis that appears on this line. I fail to see where it begins.

Answer: Thanks for pointing out the flaw. We removed the unnecessary parenthesis.

L102. Please indicate what is the meaning of the line that connect the different countries.

Answer: Thanks for pointing out this flaw. We added the following information regarding the current research focus and trends in section 2: “Most of the articles describe the effects of different types of cellulose and emulsification procedures followed by a characterization of the emulsion stability.” In the conclusions, we added: “Current research focus and trends focus mainly on material science and engineering issues. Although this is important to create new products, there is a need...”

The missing explanation about the picture is: “In this picture, the darker the blue hue, the higher the number of publications per country. Furthermore, the thicker the red lines, the higher the number of publications with coauthors from the connected countries, showing a higher interaction between countries in the research networks.”

L110-111. Please connect these sections, they appear separated and the message that is trying to convey might get lost.

Answer: Thanks for the advice. We added a link to indicate the importance of involving more countries on the topics discussed on sections 3, 4, and 5: “Developing countries and regions could use the results from previous research on nanocellulose for Pickering emulsion in basic and applied studies. Furthermore, scientists from these countries should be involved in future investigations of the effects of the Pickering emulsions and their components on the environment and human health.”

Nanocellulose for Pickering emulsions

L122-126. I think that it is important to mention in this paragraph that the surfaces of CNC can be modified according to the isolation reagents used, altering its physio-chemical behavior.

Answer: Thanks for the information. We wrote the example of how different types of acids yield different types of cellulose nanocrystals: “The surfaces of CNC can be modified according to the isolation reagents used. For example, hydrolysis with sulfuric acid adds negative charges to the surface, improving the zeta potential of the nanocrystals. Nevertheless, hydrochloric acid hydrolysis breaks the cellulose without changing the crystalline surface [18].”

L124-125,130. Please include a size range to more accurately describe the morphology of the CNC and CNF.

Answer: Thanks for indicating an important missing information. We added dimensions for CNC width (5 – 20 nm) and length (100 - 300 nm), and CNF width (25 – 70 nm).

L132-138. Please revise the clarity of this paragraph, as the writing in not clear enough.

Answer: The paragraph was rewritten to improve clarity: “Cellulose presents advantages over other biomacromolecules for Pickering emulsion preparation. For example, starch is an isomer of cellulose, but it is much more hydrophilic than cellulose. Polar liquids wet starch, but this biopolymer cannot be wetted by nonpolar liquids unless it is chemically modified or gelatinized [22]. Therefore, nanocellulose structures may not need mandatory surface modification, as starch nanostructures. Chitin may modulate immune responses [23] and chitosan requires the deacetylation modification of chitin before it is suitable to stabilize Pickering emulsions [24], which are short-comings of both biopolymers in comparison to cellulose. Proteins may be denatured by food processing conditions, like temperature or pH [25], making them unsuitable as Pickering systems stabilizers. CNC and CNF may be extracted at higher temperatures and acidic conditions, making them viable options for Pickering emulsions in food products that proteins may not resist as proper additives.”

L154. Please provide a reference.

Answer: The text flow had a flaw and the information that should in one paragraph was split into two, causing the problem observed by the reviewer. We merged the paragraphs, and the sentence without reference is closer to the next sentence. The information from both sentences come from reference [28], fixing the problem.

L182-183. It is important to identify the criteria used to mention that the Pickering emulsifying stability was better, and include the data to support this affirmation.

Answer: The sentence was poorly written. We rewrote it and add supporting information: “Acid-extracted CNCs from pineapple peels with different contents of residual lignin were used to prepare oil-in-water Pickering emulsions with commercial sunflower oil. After 15 days of storage at 4°C, the systems with 11% lignin on the CNCs presented smaller droplets than emulsions prepared with 5% lignin on the CNCs. The emulsion prepared with the higher bleached cellulose (lower lignin content) also had a water layer at the bottom of the flasks, indicating less emulsion stability [31].” Unfortunately, the authors did not measure the size of the droplets, and all the discussion is based on striking differences on the pictures.

L187-188. Please provide a value of the reduction observed in the droplet size of the lignocellulosic nanofibrils with high lignin, compared to the ones with a lower lignin content.

Answer: Unfortunately, we cannot provide this reduction in a quantitative way. The authors of the article (https://doi.org/10.1016/j.foodres.2021.110738) did not measure the size of the droplets or released the original data, and all the discussion is based on striking differences on the pictures.

L189. I suggest to reconsider the name of this section. Perhaps, effect of the phase composition in the stability of Pickering emulsions or Impact of the phase composition in the stability of Pickering emulsions.

Answer: Thanks for the advice. We changed the section to “Impact of the phase composition on the stability of Pickering emulsions.”

L190-192. It is unclear the type of oils evaluated, and the effect of other characteristics of the oil phase, such as chain size and polarity of the oil or mixture of oils. This section shall be improved with more in-depth information.

Answer: Thanks for the comment. We included more details about the publication to increase the clarity of the paragraph: “A wide set of different types of oil (corn oil, sunflower oil, flaxseed oil, orange oil, Miglyol® 812N, and fish oil) was tested under the same processing conditions to prepare oil-in-water emulsions stabilized by CNC extracted from wood by sulfuric hydrolysis. Orange oil had the smallest density, 0.84 g/cm3, while the density of the other oils ranged from 0.92 to 0.95 g/cm3. The emulsion with orange oil presented the largest droplet size. The droplet size of the emulsion prepared with flaxseed oil was statistically (p<0.05) smaller than the other emulsions, even though flaxseed oil had similar characteristics of triglyceride composition [32].”

We wished we could use values to indicate how small are the flaxseed oil droplets, but the authors did not provide this information or the original data.

L209-L213. According to this paragraph, it would be valuable to mention if there is an effect of the pH and the presence of other components, such as proteins, surfactants or other carbohydrates in the emulsifying properties of nanocellulose.

Answer: Thanks for the valuable contribution. We changed the last sentence: “Additionally, researchers must pay closer attention to other components and characteristics of the systems, such as proteins, surfactants, or other carbohydrates in the emulsifying properties of nanocellulose. These variables, not usually accounted on most of experiments, may alter the interaction between the nanostructures and the phases. For example, changes in the pH may alter the surface charge, inducing coalescence of droplets [33].”

Food application of Pickering emulsions

L216-220. Please indicate an approximate number of patents, to reinforce this paragraph.

Answer: Thanks for the comment. We ran a search on the Google Patents website and found around 280 patents. The paragraph was changed to include the information about the number of publications and number of patents: “One of the main applications of Pickering emulsions is the production of emulsion-based food products. There are many articles (more than 650 documents) relating to biopolymers (Scientometrics analysis section), including cellulose nanostructures, for this application. There are also patents about equipment and procedures to prepare Pickering emulsions for food applications [36,37]. In a recent search on the Google Patents website (Sep. 6th, 2023), almost 280 patents were found with the terms “nanocellulose” and “Pickering”.”

L222-225. It is important to mention that HPMC and Na-CMC are stabilizers to improve the stability of the emulsion.

Answer: Thanks for the comment. We added this discussion after the reference [38]: “These results indicated that HPMC and Na-CMC improved the stability of the emulsion.”

L228. Please include the range, and, if possible, the time the emulsion remained stable or the increase in this time observed in CNC and sorbitan monostereate.

Answer: Thanks for pointing out this flaw. We provided the requested information and the stability indices for the different systems: “The stability indices (percentage ratio of the volume of the emulsion layer and total volume) after 72 hours of the systems prepared with sorbitan were between 2.99 and 4.37 no matter the tested pH (3, 7, or 11) or temperature (2°C or 25°C), while the stability indices of the Pickering emulsions were always above 97% in all tested conditions [39].”

L229. It is not clear which property is imperative.

Answer: Thanks for indicating a missing information. We rewrote the sentence to make clear the example of higher stability achieved with nanocellulose than traditional surfactants: “The authors point out that the stability of an emulsion in a large range of pH values and temperatures is imperative when producing a food product. For example, a sauce may contain acidic additives such as organic acids or citric juices and may be stored at room temperature (≈25°C) or refrigerated (≈4°C). In the case of this product, a cellulose-based Pickering emulsion may be more efficient than a system prepared with usual surfactants.”

L237. Please provide a reference of the conditions that are relevant during the processing through the gastrointestinal tract.

Answer: We described the chemical conditions in the human stomach and added a reference: “Cellulose is non-digestible by the enzymes in the upper part of the human gastrointestinal tract, an acidic environment with proteolytic enzymes, therefore nanocellulose materials appear as promising structures to stabilize emulsions [41].”

L240. Please provide the percentage of digestibility reduction.

Answer: Thanks for the comment. We added the requested information and a missing reference. The sentence was rewritten as: “A Pickering oil-in-water emulsion prepared with wood CNC reduced the simulated small intestine digestibility of corn oil from ≈60% to ≈20% [41].” Additionally, we adjusted the numbering of all further references.

L242. I am not sure if method is the most appropriate word here, maybe approach or application could be better in this situation.

Answer: The term was changed from “method” to “application.”

L251-252. It is not custom to use the question mark in the paragraph. The ethical dilemma can be highlighted by other means.

Answer: The sentence was rewritten: “Another point to be discussed is the ethical dilemma of preventing oil absorption in some foodstuffs while there is a shortage of calories for people in many countries.”

269-272. While packaging and food packaging are a very active subject among the applications of nanocellulose, maybe, the use of Pickering emulsions as precursors might not be as active. It would be interesting to complement this paragraph with the results obtained in the Scientometric analysis.

Answer: Thanks for the suggestion. We reanalyzed the database and discovered that only 31 in 655 articles are related to Pickering emulsions and packaging. We added a discussion to the paragraph: “Although the research on this subject is exciting, more scientists should work on this topic. Only 31 articles from the database of 655 retrieved articles deal with potential applications of Pickering emulsions for packaging.”

L272. I think these lines shall be revised, as the comparison is not clear, and does not necessarily hold true, as it depends on which material is being compared and what is understood by “efficient”, is the efficiency related to the processing of the material or is it related to the barrier performance.

Answer: Thanks for the comment. We removed the last sentence of the first paragraph and merged the first and second paragraphs. We agree that “efficiency” is a very complex topic and the section should not begin with this type of ambiguity.

L281. Please provide a reference.

Answer: The missing reference was added: [43].

L301. Please indicate the main result obtained in that article, if possible, indicating the thermal conductivity or other measure of the thermal isolation provided by the packaging material.

Answer: We added the requested information, improving the clarity of the results obtained with the paper with paraffin droplets: “The PCM paper had a total enthalpy of 139 J/g between 15°C and 35°C, close to the value for pure paraffin (183 J/g). Cycles of heating and cooling down indicated that the PCM paper survived for 100 cycles without paraffin leakage. A mass of 2 g of PCM paper was applied to a surface of 25 cm2 and heated with a simulated sun with an intensity of 1000 W/m2. The temperature under the surface with no covering changed from 25°C to 35°C in 5 minutes, while covering the surface with the PCM paper increased the time to 23 minutes [47].”

L317-318. I am wondering if this application should be included in the section pertaining to packaging materials, also it would be interesting to briefly mention the results obtained with these packages.

Answer: We reformulated the writing to avoid confusion and improve the flow, removing a discussion about package in the section of innovative food applications. We presented the findings of the article and fixed the writing of our discussion, making clear that the authors created a structure that can be used to grow cells and make artificial meat: “A Pickering emulsion stabilized with CNC was used as 3D printing ink [55] to prepare custom-made structures. The authors created oil-in-water Pickering emulsions with 80% oil, stabilized with 0.5 wt% of CNC, suitable for 3D printing, even with the ionic strength of 50 mM NaCl. The SEM of the printed materials is a scaffold of apparent high surface area [55]. This 3D-printed sponge may potentially be applied to grow cells, developing novel 3D food products.”

Sustainability of Pickering emulsions

L361-362. Please maintain the format of the previous steps.

Answer: The list was fixed to maintain the format of the previous steps.

L393-400. There is a coherence issue with these lines. The authors indicate that oral evaluations have been conducted, however, they proceed to explain the effect of nanocellulose inhalation in lung cells.

Answer: Thanks for indicating the mistake. We corrected from “oral” to “oropharyngeal.”

L418. Please check this line, it is not clear.

Answer: The sentences were rewritten: “The diversity of types of cellulose nanoparticles has delayed the elaboration of safety profiles that potential hazards with their characteristics because of the plethora of different widths, lengths, surface charges, residual lignin and hemicellulose, for example [69].”

L431. Please verify the redaction of this line.

Answer: Thanks for your comment. We rewrote the sentence to clarify the meaning: “An organization has to ask the FDA to accept a new chemical as GRAS and provide enough evidence. A committee in the FDA will review and evaluate the presented documents before deciding to include a new chemical in the GRAS list.”

L433. Please provide a reference. Also, it is important to keep in mind, that ethylcellulose is approved as a food additive in only three cases, as indicated by the FDA (https://www.accessdata.fda.gov/scripts/cdrh/cfdocs/cfcfr/CFRSearch.cfm?fr=172.868).

Answer: Thanks for your contribution. We split the references at the end of the paragraph, moving the reference [72] closer to the sentence. We made clear the information of the use of ethylcellulose as food additive in the text: “An organization has to ask the FDA to accept a new chemical as GRAS and provide enough evidence. A committee in the FDA will review and evaluate the presented documents before deciding to include a new chemical in the GRAS list. For example, ethyl cellulose has been recognized as GRAS since 2013, when the FDA analyzed the claim from Dow Wolff Cellulosics [72]. Even in this case, ethyl cellulose is approved as a food additive in only three situations. No organization requested the FDA to analyze hydroxypropyl cellulose as GRAS, and this chemical is not listed as GRAS [73].”

L542. Please indicate what does ILCD/PEF stands for.

Answer: Thanks for the comment. We explained the meaning of the abbreviations: “(International Life Cycle Data System/Product Environmental Footprint).”

Conclusions

The section about sustainability of Pickering emulsions could be mentioned more in this section, for example, the existence of the LCA of oils and cellulose is not mentioned.

It would be interesting to specify the type of research required to improve the calculation of the environmental and social impacts for the incorporation of Pickering emulsions involving nanocellulose, considering the intricate complexities of this subject.

Answer: Thank you very much for this important feedback. We added the requested discussion: “There are publications of life cycle analysis of the components of the emulsions but not of the emulsions themselves. More interaction is needed among health, material sciences, and food engineering researchers to address the complex issues regarding approving a food containing Pickering emulsions. Ultimately, legal and communication professionals must be involved in the discussion of new food products to prevent public backlash and government misunderstanding, especially when these products have reached the technological maturity required for commercialization.”

Comments on the Quality of English Language

The use of the language through the text could be improved, there are cases of awkward phrasing and, more importantly, some sentences are difficult to read, which reduces the comprehensibility of the message they are trying to convey. 

It is important to verify the language and make the changes accordingly.

Answer: Thanks for your comment. We revised the document and used a cloud-based typing assistant. We hope the quality of the English writing improved, increasing the comprehensibility of the messages we are conveying.

Reviewer 2 Report

In this manuscript, the authors reviewed the nanocellulosed-based Pickering emulsion which is applied in food for sustainable goals. They summarize the mechanisms, applications, and environmental impacts and provide opinions on future potential topics for future research. Some minor concerns are listed as follows:

1.     This review is entitled “Food Emulsion”, but is entirely focused on Pickering emulsion. Why not give a specific title with “Pickering emulsion” or review the regular cellulose-based regular emulsion? 

2.     Figure 1 Page 3, What does the different colour in the map stand for? It is also mentioned in the conclusion part that “The research is still concentrated in a few countries, especially developed and emerging countries, with low participation from Asiatic and African nations.” Could the authors summarize the research focus and trend in different areas? 

3.     Section 3 Pages 3-6, How is the emulsifying ability and stability difference among CNF, CNC, and BC. Which one can provide better anti-coarsen and anti-disproportionate performance in food emulsion stabilization?

4.     Paragraph 3 Section 3 Page 3, why shorter CNC resulted in smaller droplets.

5.     How is the application of nanocellulose in w/o type Pickering emulsion?

6.     Section 4.2 Pages 7-8, some questions were not clearly stated 

a.     What forms of food packaging materials are needed to be produced via Pickering emulsion?

b.     Which type of emulsions are needed? o/w or w/o?

c.      What is the role of polymer (PMMA, PLA, PS) in the nanocellulose based Pickering emulsion? What is the role of polymer (PMMA, PLA, PS) in the nanocellulose-based Pickering emulsion?

The manuscript was well written and organised.

Author Response

Review Report Form – Reviewer 2

In this manuscript, the authors reviewed the nanocellulosed-based Pickering emulsion which is applied in food for sustainable goals. They summarize the mechanisms, applications, and environmental impacts and provide opinions on future potential topics for future research. Some minor concerns are listed as follows:

We thank the reviewer for providing important feedback to improve the document.

1. This review is entitled “Food Emulsion”, but is entirely focused on Pickering emulsion. Why not give a specific title with “Pickering emulsion” or review the regular cellulose-based regular emulsion?

Answer: Accepted

2. Figure 1 Page 3, What does the different colour in the map stand for? It is also mentioned in the conclusion part that “The research is still concentrated in a few countries, especially developed and emerging countries, with low participation from Asiatic and African nations.” Could the authors summarize the research focus and trend in different areas?

Answer: Thanks for pointing out this flaw. We added the following information regarding the current research focus and trends in section 2: “Most of the articles describe the effects of different types of cellulose and emulsification procedures followed by a characterization of the emulsion stability.” In the conclusions, we added: “Current research focus and trends focus mainly on material science and engineering issues. Although this is important to create new products, there is a need for the development of scaling-up technologies, allowing the production of kilograms or liters per hour of the products.”

The missing explanation about the picture is: “In this picture, the darker the blue hue, the higher the number of publications per country. Furthermore, the thicker the red lines, the higher the number of publications with coauthors from the connected countries, showing a higher interaction between countries in the research networks.”

3. Section 3 Pages 3-6, How is the emulsifying ability and stability difference among CNF, CNC, and BC. Which one can provide better anti-coarsen and anti-disproportionate performance in food emulsion stabilization?

Answer: Thanks for indicating the missing information. It should have been added to the end of section 3.1. We added the following paragraph: “The process of nanocellulose extraction impacts the efficiency of the Pickering emulsion stability. Researchers and engineers should evaluate the most suitable options for their planned products considering economic, environmental, and operational aspects. For example, using CNC from rice husks may be a fitting option in Asia, while CNF from birch may be an affordable choice in Europe.”

4. Paragraph 3 Section 3 Page 3, why shorter CNC resulted in smaller droplets.

Answer: Thanks for the comment. We added the missing explanation: “Indeed, shorter CNCs resulted in smaller droplets [28] and have a better efficiency in stabilizing Pickering emulsions because these shorter nanostructures have a higher packaging density, covering a higher surface area of the droplets, increasing steric obstruction between the droplets, resulting in a higher number of smaller oily structures [4].”

5. How is the application of nanocellulose in w/o type Pickering emulsion?

Answer: Thanks for showing us this missing information. We added information from reference [4] in the section 3.1.: “If the hydrophobicity increases in the nanocellulose, water-in-oil emulsions are prepared, and if the hydrophilicity increases, oil-in-water systems are formed [4].”

6. Section 4.2 Pages 7-8, some questions were not clearly stated

a. What forms of food packaging materials are needed to be produced via Pickering emulsion?

Answer: We added examples of forms of food packaging that can be prepared via Pickering emulsion with the following sentence: “Articles present exciting results of how Pickering emulsions can be used to prepare or functionalize packaging materials, such as films or foams with improved mechanical, thermal, and barrier characteristics [43,44,45–49].”

b. Which type of emulsions are needed? o/w or w/o?

Answer: We explained in all cited examples in the manuscript the type of emulsions. In all cases, they were o/w emulsions.

c. What is the role of polymer (PMMA, PLA, PS) in the nanocellulose based Pickering emulsion? What is the role of polymer (PMMA, PLA, PS) in the nanocellulose-based Pickering emulsion?

Answer: We explained the role of the polymers in all cited examples in the manuscript. It is clearer now which polymers were used for film, foam, or capsules.
